# Lignin Nanoparticle-Coated Celgard Separator for High-Performance Lithium–Sulfur Batteries

**DOI:** 10.3390/polym11121946

**Published:** 2019-11-27

**Authors:** Zengyao Zhang, Shun Yi, Yuejia Wei, Huiyang Bian, Ruibin Wang, Yonggang Min

**Affiliations:** 1School of Materials and Energy, Center of Emerging Material and Technology, Guangdong University of Technology, Guangzhou 510006, China; gdjy_zzy@163.com (Z.Z.); 13290065303@163.com (S.Y.); wyj20190726@163.com (Y.W.); 2Jiangsu Provincial Key Laboratory of Pulp and Paper Science and Technology, Nanjing Forestry University, Nanjing 210037, China; hybian1992@njfu.edu.cn

**Keywords:** lignin nanoparticles, separator, Li–S batteries, biomass, acid hydrotrope

## Abstract

Tremendous efforts have been made toward the development of lithium–sulfur (Li–S) batteries as one of the most reasonable solutions to the rapidly increasing demand for portable electronic devices and electric vehicles, owing to their high cost-efficiency and theoretical energy density. However, the shuttle effect caused by soluble polysulfides is generally considered to be an insurmountable challenge, which can significantly reduce the battery lifecycle and sulfur utilization. Here, we report a lignin nanoparticle-coated Celgard (LC) separator to alleviate this problem. The LC separator enables abundant electron-donating groups and is expected to induce chemical binding of polysulfides to hinder the shuttle effect. When a sulfur-containing commercially available acetylene black (approximately 73.8 wt% sulfur content) was used as the cathode without modification, the Li–S battery with the LC separator presented much enhanced cycling stability over that with the Celgard separator for over 500 cycles at a current density of 1 C. The strategy demonstrated in this study is expected to provide more possibilities for the utilization of low-cost biomass-derived nanomaterials as separators for high-performance Li–S batteries.

## 1. Introduction

With the drastic growth of portable electronic devices and electric vehicles, the interest on rechargeable energy storage systems with high energy density, high power density, and long lifetime [1,2,3] has rapidly increased. Among the available options, lithium–sulfur (Li–S) batteries have been regarded as one of the most promising energy storage devices in the near future due to their high specific energy (2500 Wh·kg^−1^) and theoretical specific capacity (1675 mAh·g^−1^). However, the commercialization of Li–S batteries is still hindered by many technical challenges such as the shuttle effect caused by soluble polysulfides [4], volume expansion of sulfur during the charge/discharge process [5], low electrical conductivity of sulfur, etc. [6,7,8]. To facilitate the development of the Li–S system, it is therefore crucial to alleviate the shuttle effect and improve the conductivity of the sulfur cathode.

In recent years, impregnating sulfur into tailored porous/hierarchical/nano-architectured carbon matrixes [9,10,11] and compositing conducting polymers (such as polyaniline [12], poly(N-methylpyrrole) [13], etc.) or metal organic framework/metal sulfides (oxide) [14,15,16,17,18] are the mainstream solutions to addressing these issues. Although these techniques can provide valuable insights into exploring high-performance Li–S batteries with impressive capacity and cycling stability, the use of expensive components/equipment and complex operations remain the major bottlenecks to wide utilization.

In this context, rational design of the structure and/or composition of separators is recognized as one of the most efficient solutions. Recently, intensive efforts have focused on modifying the commercial porous membranes (e.g., polypropylene [2,19,20], polyethylene [21,22], Nafion [23,24], glass fibers [25], etc. [26]) to provide new perspectives. These composite separators can serve as the electronic insulator among electrodes and influence the transportation of polysulfides. However, although these studies have shown pronounced improvements, some crucial issues remain to be resolved: (1) most ingredients used for the fabrication of the above separators are non-renewable; (2) energy-intensive processes, such as high-temperature carbonization and electrospinning, are included. Developing a low-cost and renewable biomass-modified separator therefore seems to be urgent. Inspired by the blossoming f research activities centered on the continued push for cheaper and more sustainable chemistries of biomass processing [27,28,29], low-cost lignin that can be greenly processed is expected to be a promising reinforcing agent for the separator of Li–S batteries.

Here, we report a simple strategy using a lignin nanoparticle (LNP)-coated Celgard (LC) membrane as a novel separator for Li–S batteries. The low cost and abundant quinone of lignin engender a versatility for its application in diverse technologies such as sensors [30,31], solar cells [32], supercapacitors [33,34], and lithium-ion/sodium-ion batteries [35,36]. Thanks to their good dispersity [28,29], LNPs are easily deposited on a conventional Celgard separator by simple filtration to obtain a composite separator. When used in Li–S batteries, LNPs can chemically alleviate the polysulfides’ diffusion, thereby significantly suppressing the polysulfides’ shuttle, resulting in ultrahigh coulombic efficiency. Meanwhile, the quinone in LNPs can absorb electrons and release protons to facilitate the transport of lithium ions which guarantees a long lifecycle and good rate capability of the Li–S battery. These encouraging results pave the way for a viable conversion of biomass-derived lignin into a sustainable and low-cost functional material for energy storage.

## 2. Materials and Methods

### 2.1. Preparation of Lignin Nanoparticles (LNPs)

Lignin nanoparticles were produced according to previously reported procedures [37,38] in which the feedstock and acid hydrotrope were aspen wood sawdust and benzenesulfonic acid (B-acid), respectively. Briefly, 40 g of B-acid was mixed with 10 g of deionized (DI) water in a 100 mL flask. Then the mixture was heated to 80 °C under mechanical stirring (200 rpm) until a clear solution was achieved, followed by the addition of 2.5 g (oven-dry weight) of cypress sawdust. The reaction was maintained at 80 °C for 20 min under mechanical stirring (500 rpm). At the end of the reaction, the flask was removed from the heat source and the resultant mixture was vacuum filtrated immediately. The filtrate was dialyzed against DI water for 3 days and then oven-dried at 60 °C for 12 h to obtain LNPs.

### 2.2. Fabrication of LNP-Coated Celgard (LC) Separator

The LC separator was fabricated according to Tang’s [21] study. In brief, LNPs were first dispersed in ethanol (80% (*v*/*v*), pH value was adjusted to 9 by adding a desired amount of NaOH) under magnetic stirring until a clear solution was obtained. Second, the resultant solution was subjected to vacuum filtration using the Celgard membrane (Celgard 2400, Celgard Company, Charlotte, NC, USA). Subsequently, the filter cake was vacuum-dried at 50 °C overnight with Celgard to give an LC separator of approximately 0.72 g·cm^−2^ at the LNP loading. The facile fabrication of the LC separator is schematically illustrated in Figure 1. In addition, to investigate the adhesion of LNPs on the LC separator, a peeling test was utilized according to previous research [39,40] and the results indicated that LNPs have good adhesion on the Celgard substrate.

### 2.3. Characterizations

The Fourier transformed infrared (FTIR) spectrum of the LNPs was obtained on a commercial FTIR spectrometer (Nicolet 6700, Nicolet Instrument Co., Pittsburgh, PA, USA) over the wavenumber range of 4000–400 cm^−1^. A simultaneous thermogravimetric analysis (TGA)/differential scanning calorimetry (DSC) analyzer (TGA/DSC 3+, Mettler Toledo, Greifensee, Switzerland) was used to qualitatively analyze LNPs and quantitatively determine the sulfur content of the cathode. In detail, all samples were heated to 800 °C at a heating rate of 10 °C·min^−1^ under N_2_ atmosphere of 20 mL·min^−1^. The morphology and pore structure of the Celgard membrane and the LC separator were observed by a Hitachi SU8010 field emission scanning electron microscope (FESEM, Hitachi Co., Tokyo, Japan).

### 2.4. Electrochemical Measurements

For the electrochemical measurement, each cathode was prepared according to the method reported by Cai et al. [6], which is commonly used for Li–S batteries. In brief, sulfur powder was mixed with acetylene black and polyvinylidene fluoride (weight ratio of 8:1:1) in nmethyl-2-pyrrolidene under vigorous agitation. The mixed slurry was then moved onto a clean aluminum foil and vacuum dried at 60 °C for 12 h.

All cathode and separator samples were cut 12 and 16 mm in diameter, respectively. A standard cell assembly process for 2032 coin cells was carried out in an argon-filled glove box with a pair of cathode and metallic lithium anode separated by either LC separator or Celgard soaked with 20 μL of electrolyte. Therefore, the amount of the electrolyte introduced to the cell corresponds to 23 mL of electrolyte per gram of sulfur. In addition, the electrolyte used in this study was of 1,3-dioxolane, and 1,2-dimethoxyethane 1:1 (*v*/*v*) with bis-(trifluoromethane) sulfonamide lithium (1.0 M) and 1% LiNO_3_.

The cells with different separators were galvanostatically discharged/charged between 1.7–2.8 V (VS Li/Li^+^) at room temperature on a multichannel potentiostat system (Neware Battery Co., Shenzhen, China). The cyclic voltammetric and electrochemical impedance spectroscopy (EIS) measurements were performed on an electrochemical workstation (CHI760e, Chenhua Instruments, Shanghai, China). The EIS measurements were conducted at a frequency ranging from 10^−2^ to 10^5^ Hz at the AC amplitude of 5 mV and an open circuit voltage. The specific capacity of the electrodes was calculated on the basis of the total weight of sulfur determined by TGA which was approximately 1.5 mg·cm^−2^.

## 3. Results and Discussion

### 3.1. Structure of LNPs

The FTIR was carried out to qualitatively confirm the presence of lignin in LNPs, and the peak assignments were conducted according to the literature [37,41]. As shown in Figure 2a, the bands at 1610, 1510, and 1465 cm^−1^ are ascribed to C=C benzene ring vibration, aromatic skeletal vibration, and asymmetric bending in CH_3_ of lignin, respectively. Though weaker than the ester group studied in Goodenough’s [42] paper, the C=C benzene ring and CH_3_ on LNPs were still of electron-donating groups and, thus, were expected to bind polysulfides to alleviate the shuttle effect. Figure 2b shows that less than 2% of the weight of LNPs was lost in the range of 100–200 °C which indicates that the LNPs included very few volatiles. Also, a weight loss of about 37% that is attributed to the release of aromatics, carbonyls, alkyls, CO_2_, and CO was observed in the range of 200–500 °C. Further, the characteristic feature that is ascribed to aspen-derived lignin between 200–500 °C [43] was observed as a weight loss peak at 384 °C in the corresponding DTG curve in Figure 2.

### 3.2. Pore Structures of Separators

The FESEM images of pure Celgard and the LC separator are displayed in Figure 3a,b. It is clear that pure Celgard shows a macroporous arrangement in a representative elliptic shape (highlighted in Figure 3b) of an average semi-major axis/semi-minor axis of approximately 100/33 nm. For the LC separator, LNPs with an average diameter over 100 nm were found deposited on most macropores (Figure 3c,d). The latter is consistent with the fact that larger pores can retain more feedstocks which is common sense in the papermaking industry. Combining this with the above FTIR results, the electron-donating groups of LNPs on the LC separator were expected to induce chemical binding of polysulfides and restrict the polysulfides on the cathode. For a proof of concept, the diffusion behavior of polysulfides was observed in an H-type cell configuration with each of the two separators. As shown in Figure 3e–f, the cell with the Celgard separator underwent much severer polysulfides permeation which displayed a dark yellow color in the right chamber (the right unit in Figure 3f) after 24 h in contrast to the LC separator, half of whose right chamber (the left unit in Figure 3f) well maintained a clear and transparent appearance. That is, the LC-separator may comparably offer an enhanced capability to restrain the diffusion of polysulfides across it. In addition, the LC separator showed an unobvious immersion–height difference with respect to the Celgard separator (Figure 5g). This can be understood by the fact that LNPs coated on Celgard have no effect on its porosity (Figure 3a–d); thus, it has good electrolyte uptake performance [44].

### 3.3. Electrochemical and Long-Term Cycling Stability

As shown by the typical CV curve at a scan rate of 0.1 mV·s^−1^ (Figure 4a), the reduction scan was composed of two cathodic peaks at 2.2 and 1.9 V, representing reduction of the elemental sulfur to soluble high-order polysulfides (Li_2_S_n_, 4 ≤ n ≤ 8) and their further reduction to solid lithium sulfides (Li_2_S_2_/Li_2_S), respectively. This is much different from that of the neat Celgard separator (shown in Figure 4b). In the subsequent anodic scan, one oxidation peak at 2.5 V was discerned that related to the conversion of polysulfides to elemental sulfur [21,45] with facile electrochemical kinetics. As determined from the TGA curves presented in Figure 4c, a sharp weight loss was observed between 200 and 300 °C, corresponding to the evaporation of sulfur in the composite. That is, the sulfur content of the cells used in this study was estimated to be 73.8 wt % on the basis of which all the specific capacities of both separators addressed in the following were calculated. The rate performances with the LC separator and the Celgard separator were evaluated at various current rates from 0.1 C to 2 C (Figure 4d). The cell armed with the Celgard separator suffered from dramatic capacity decay (the bottom curve) of which the specific capacity sharply decreased from the initial 167 mAh·g^−1^ (100%) to a final value of 140 mAh·g^−1^ (83.8%). In contrast, the LC separator demonstrated much better performance (the middle curve) and the observed specific capacity only changed from the initial 380 mAh·g^−1^ (228%) to the final 377 mAh·g^−1^ (226%). Moreover, the LC separator could deliver initial capacities of 1006 (0.1 C), 777 (0.2 C), 603 (0.5 C), and 487 (1 C) mAh·g^−1^, outperforming 937 (0.1 C), 743 (0.2 C), 558 (0.5 C), and 424 (1 C) mAh·g^−1^ of the Celgard separator.

### 3.4. Long-Term Cycling Stability

For comparison purposes, the cycling stability of the LC separator and the Celgard separator was assessed at a current density of 1 C. Figure 5a shows that two cells delivered close initial specific capacities (∼450 mAh·g^−1^) and good coulombic efficiencies (>99%). However, the capacity of the Celgard separator presented a significant 64.9% decrease in the subsequent 500 cycles, while this feature for the LC separator was only 34.8%. The SEM images reveal that numerous macropores and mesopores existed in the Celgard separator, which was useless in retarding the polysulfides loss and, consequently, led to its poor cycling stability. In contrast, as confirmed above, the LC separator coated with LNPs could trap the polysulfides through chemical binding. The long-term cycling stability of the LC separator was further investigated by evaluating the evolution of over 660 charge/discharge cycles (Figure 5b). The first two cycles were performed at 0.1 C, and the subsequent 660 cycles were tested at 1 C. The initial discharge capacity of the LC separator was 405 mAh·g^−1^ at 1 C, and its discharge capacity slowly decreased and ultimately reached 272 mAh·g^−1^ after 660 cycles, which corresponded to a 0.168% capacity decay per cycle over 660 cycles. In most cases, the LC separator achieved a coulombic efficiency of over 99% which implies an excellent electrochemical reversibility. Figure 5c,d presents several typical galvanostatic charge/discharge curves of both separators in long-term cycling at 1 C which consisted of two reduction plateaus and one long oxidation plateau, representing the redox reactions of a typical Li–S battery. More information about capacity retention was obtained from these cycles. The capacity retentions of the LC separator were 100.8% after 100 cycles, 77.7% after 300 cycles, and 72.9% after 400 cycles, whereas the corresponding capacity retentions of the Celgard separator were 69.0%, 46.0%, and 40.1%. These results are consistent with the rate performance determination, indicating that the overwhelming cycling stability of the LC separator over the Celgard separator was maintained even at a high rate, especially considering that the discharge plateau voltage of the LC separator stabilized around ~2.0 V. The EIS measurements were conducted to further demonstrate the electron and ion transfer capability of these separators. As shown in Figure 5e, the charge transfer resistance (R_ct_, corresponding to the diameter of the semicircle at high frequencies) was 27 and 87 Ω for the LC separator and Celgard separator, respectively. The smaller R_ct_ for the LC separator suggests a fast lithium ion diffusion in the LC separator which might be attributed to the electron-donating groups of LNPs on the LC separator that can induce chemical binding of polysulfides and restrict the polysulfides on the cathode, ensuring their better contact than the Celgard separator. Moreover, it was found that this benefit of the LC separator would be compromised when the LNPs loading increased to 3.20 g·cm^−2^ which confirmed the above speculation, because higher LNP loading usually leads to severer agglomeration (Figure 5e). 

## 4. Conclusions

Biomass-derived lignin nanoparticles were prepared through a green acid hydrotrope method and successfully coated onto Celgard membrane. We demonstrated that a lignin nanoparticle-coated Celgard separator could alleviate the shuttle effect. On one hand, LNPs can serve as a temporary electrolyte reservoir to restrain the soluble polysulfides from directly diffusing into the bulk electrolyte. On the other hand, LNPs have abundant electron-donating groups that can induce chemical binding of polysulfidesand consequently improve the cycling stability. The strategy demonstrated in this work will open the door toward developing efficient biomass-derived separators using green chemistry for the practical applications of energy storage devices.

## Figures and Tables

**Figure 1 polymers-11-01946-f001:**
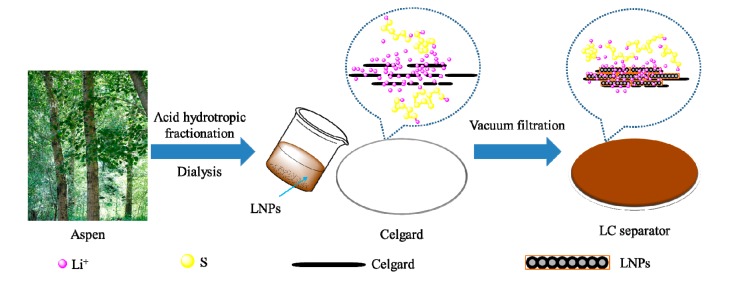
Schematic of the fabrication process to produce the lignin nanoparticle (LNP)-coated Celgard (LC) separator.

**Figure 2 polymers-11-01946-f002:**
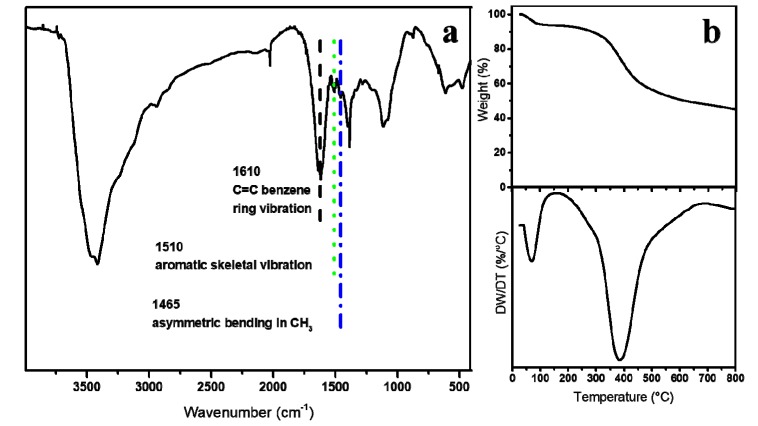
(**a**) FITR spectrum and (**b**) TGA curve of LNPs (upper) with corresponding DTG profiles (below).

**Figure 3 polymers-11-01946-f003:**
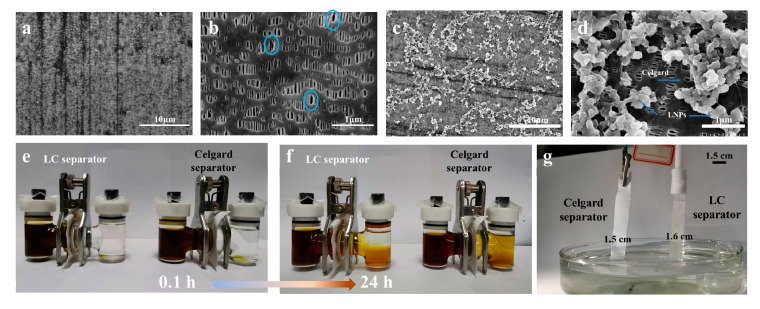
FESEM images of (**a,b**) the Celgard separator and (macropores highlighted by circles), (**c,d**) the LC separator, (**e**) optical observations of polysulfides diffusion (in both left chambers of the two units, initially 80 mg of sulfur powder and 23 mg of Li_2_S were mixed in 20 mL of electrolyte under vigorous agitation in an argon-filled glove box in advance) when equipped with the LC separator (left unit) or the Celgard separator (right unit), (**f**) 24 h after (**e**), (**g**) photograph showing electrolyte immersion–height comparison between both separators after 60 min, of which the dimension was fixed at 1.5 cm × 18 cm.

**Figure 4 polymers-11-01946-f004:**
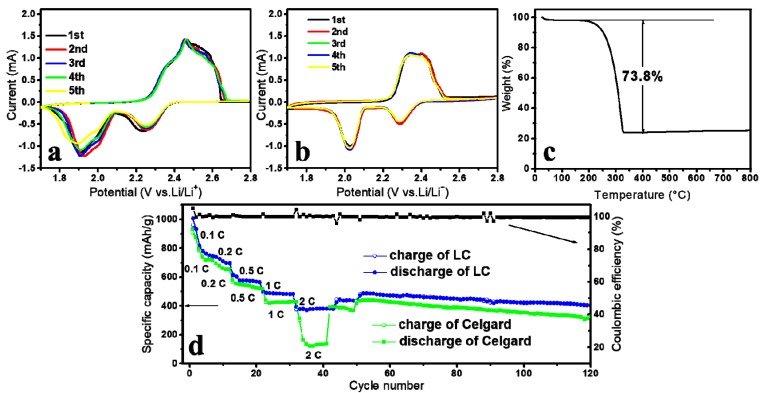
Cyclic voltammogram profile for Li–S battery with (**a**) the LC separator or (**b**) the Celgard separator at the scan rate of 0.1 mV·s^−1^. (**c**) TGA curve of the cathode utilized in this study. (**d**) Rate performance at various current rates (0.1, 0.2, 0.5, 1, and 2 C) with the LC separator (upper curves) or the Celgard separator (lower curves).

**Figure 5 polymers-11-01946-f005:**
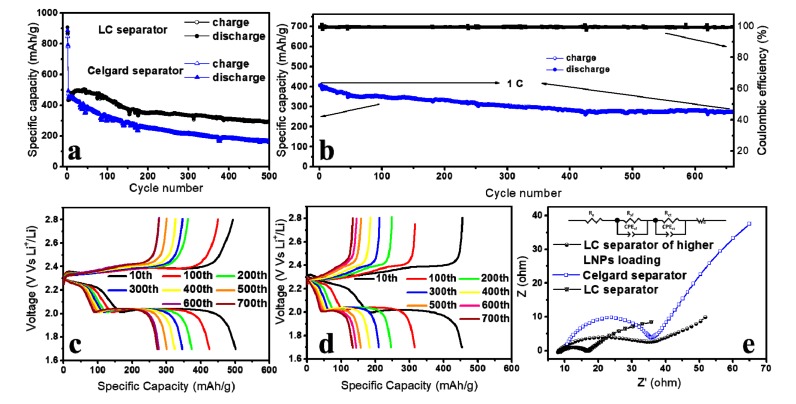
(**a**) Cycling stability comparison between the LC separator and the Celgard separator at a current rate of 1 C; (**b**) cycling stability and coulombic efficiency at a current rate of 1 C over 660 cycles with the LC separator; galvanostatic charge/discharge curves of (**c**) the LC separator and (**d**) the Celgard separator, (both at the current rate of 1 C); (**e**) Nyquist plots from the EIS of fresh cells with the LC separator of higher LNP loading (3.20 g·cm^−2^), the Celgard separator, and the LC separator.

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
