# Peer review of "Lignin Nanoparticle-Coated Celgard Separator for High-Performance Lithium–Sulfur Batteries"

_polymers, 2019, doi:10.3390/polym11121946_

Round 1

Reviewer 1 Report

The authors developed a strategy to apply naturally abundant biomass derived lignin nanoparticles as the reinforcement of the commercially available Celgard membrane for Li-S battery. The LC-separator exhibits much better rate performance and cycling stability than the Celgard separator. The authors present a cost-effective and effective strategy to fabricate the separator forwarding its practical application for high power density Li-S batteries. I believe that this paper will attract considerable interest in Li-S battery community and that it is worthy to be published in Polymers, after the following comments have been carefully addressed.

The Introduction Section can be further improved and some highly related papers related to the Li-S battery and separator modification need to be cited, such as Energ. Environ. Sci. 11(2018)2560-2568, iScience 4(2018)36-43, Adv. Energy Mater. 9(2019)1900584, Energ. Environ. Sci. 12(2019)344-350, Adv. Mater. 29(2017)1702707, J Electrochem Soc 162(2015) A1624-1629. Numerical values shouldn’t have spaces between them and the unit %, which was incorrectly used in both section 3.3 and 3.4. There are some mistypes, for example, in the 6th line of Page 7, “27, and 87 Ω for the LC separator” should be “27 and 87 Ω for the LC separator”; the fonts of Figures 2 should be uniformed as other Figures… The English should be polished a little bit for smooth reading.

Author Response

The authors developed a strategy to apply naturally abundant biomass derived lignin nanoparticles as the reinforcement of the commercially available Celgard membrane for Li-S battery. The LC-separator exhibits much better rate performance and cycling stability than the Celgard separator. The authors present a cost-effective and effective strategy to fabricate the separator forwarding its practical application for high power density Li-S batteries. I believe that this paper will attract considerable interest in Li-S battery community and that it is worthy to be published in Polymers, after the following comments have been carefully addressed.

Point 1: The Introduction Section can be further improved and some highly related papers related to the Li-S battery and separator modification need to be cited, such as Energ. Environ. Sci. 11(2018)2560-2568, iScience 4(2018)36-43, Adv. Energy Mater. 9(2019)1900584, Energ. Environ. Sci. 12(2019)344-350, Adv. Mater. 29(2017)1702707, J Electrochem Soc 162(2015) A1624-1629.

Response 1: The Introduction Section is improved, and the suggested papers are also cited.

Point 2: Numerical values shouldn’t have spaces between them and the unit %, which was incorrectly used in both section 3.3 and 3.4.

Response 2: All unit % in the manuscript has been carefully checked and revised where necessary.

Point 3: There are some mistypes, for example, in the 6 line of Page 7, “27, and 87 Ω for the LC separator” should be “27 and 87 Ω for the LC separator”; the fonts of Figures 2 should be uniformed as other Figures…

Response 3: The manuscript have been re-checked and all mistypes are corrected.

Point 4: The English should be polished a little bit for smooth reading.

Response 4: The English of this manuscript is carefully polished, and the revisions are shown in red color.

Reviewer 2 Report

Language check needs to be performed, sentences are unfinished, hard to understand the claims made. Avoid fancy words, which are not often used, as it makes paper even harder to understand, especially as not all of the words are used for their proper meaning.

Reference use is not apprpriate, - please, put citation after each instance you take somebodies thouths, and not after couple of sentences or end of paragraph. Example: "Recently, intensive efforts have focused on modifying the commercial porous membranes (e.g., polypropylene, polyethylene, glass fibers) to provide new perspectives." After this sentence the citations are needed, as it is not clear which auithors had made the discussed efforts. Please, check accross whole manuscript. Also, the citation scope is rather narrow and does not present a good overview of the results and achievements in the field.

From experimental it is not clear how electrdoes were made, How much electrolyte per gram of sulfur was used, etc. Please, separate electrode preparation, cell assembly and cycling.

How thick were separators after the coating? the membrane and polymer are not sufficiently characterized.

Fig 3. It is not clear what is placed in teh H-cell. What is in teh right container and left container (electrolyte?, solvents? please give full information). Without description it is hard to judge if experiment was made in the right way.

Fig 4. CV of both separators (with and without modification) should be shown.

The capacities shown are not impressive, - any Li-S cell can cycle stably with 400 mAh/g. after 600 cycles to have only 230 mAh/g is not good. This can be achievd without any celgard modification.

To be able to publish, authors have to find a different angle to present their research as presenting this modified celgard as soemthing usefull is not acceptable. 

Author Response

Language check needs to be performed, sentences are unfinished, hard to understand the claims made. Avoid fancy words, which are not often used, as it makes paper even harder to understand, especially as not all of the words are used for their proper meaning.

Point 1: Reference use is not apprpriate, - please, put citation after each instance you take somebodies thouths, and not after couple of sentences or end of paragraph. Example: "Recently, intensive efforts have focused on modifying the commercial porous membranes (e.g., polypropylene, polyethylene, glass fibers) to provide new perspectives." After this sentence the citations are needed, as it is not clear which auithors had made the discussed efforts. Please, check accross whole manuscript. Also, the citation scope is rather narrow and does not present a good overview of the results and achievements in the field.

Response 1: This comment is very helpful. The reference use all through the manuscript is checked and revised where necessary, with which the absent citation is also added. For the citation scope, many recent literatures of close relation are cited to broaden it.

Point 2: From experimental it is not clear how electrdoes were made, How much electrolyte per gram of sulfur was used, etc. Please, separate electrode preparation, cell assembly and cycling.

Response 2: (1) For the electrochemical measurement, each cathode was prepared according to that reported by Cai et al. [6], which is commonly used for Li–S battery. In brief, sulfur powder was mixed with acetylene black and polyvinylidene fluoride (weight ratio of 8:1:1) in nmethyl-2-pyrrolidene under vigorous agitation. The mixed slurry was then moved onto a clean aluminum foil and vacuum dried at 60 °C for 12 h.;

(2) 23 mL of electrolyte per gram of sulfur was used;

(3) The electrochemical measurement is now separated from other characterizations and composed of three paragraphs respectively include cathode preparation, cell assembly and cycling.

Point 3: How thick were separators after the coating? the membrane and polymer are not sufficiently characterized.

Response 3: The thicknesses of the LC separator and the Celgard separator are 34.9 ± 2.0 μm and 25.1 ± 2.7 μm, respectively, both of which represent the average of five measurements.

Point 4: Fig 3. It is not clear what is placed in teh H-cell. What is in the right container and left container (electrolyte?, solvents? please give full information). Without description it is hard to judge if experiment was made in the right way.

Response 4: Fig. 3e is improved (shown below) and separated into two, so its new caption is correspondingly revised as “(e) optical observations of polysulfides diffusion (In both left chambers of the two units, initially 80 mg of sulfur powder and 23 mg of Li2S were mixed in 20 mL of electrolyte under vigorous agitation in an argon-filled glove box in advance) when equipped with the LC separator (left unit) or the Celgard separator (right unit), (f) 24 h later for (e)”.

Point 5: Fig 4. CV of both separators (with and without modification) should be shown.

Response 5: CV of the neat Celgard separator without modification is added in Figure 4 as b.

Point 6: The capacities shown are not impressive, - any Li-S cell can cycle stably with 400 mAh/g. after 600 cycles to have only 230 mAh/g is not good. This can be achievd without any celgard modification.

Response 6: Sorry for this misleading. Since there are no extra active materials generated from the coated LNPs on the LC separator, it cannot increase the specific capacitance. In fact, the advantage of the LC separator over the neat Celgard is based on the former is modified to offer chemical and physical adsorptions of polysulfides to improve the long-life stability. In detail, as shown in Figure 5a, the neat Celgard separator displays a poor cycling stability as 0.33 % decay per cycle (0.15 % decay per cycle during 200 – 500 cycles), while the LC separator presents only 0.06 % decay per cycle during 200 – 500 cycles.

Point 7: To be able to publish, authors have to find a different angle to present their research as presenting this modified celgard as soemthing usefull is not acceptable.

Response 7: As responded above and added in the introduction, this study is aiming to addressing the bottlenecks for the separators of Li-S battery include non-renewable starting materials and application of energy-intensive processes, by introducing a low-cost biomass that could be processed through a well-known sustainable pathway. The lignin nanoparticles-modified-Celgard separator is barely reported, while the presented results demonstrate that it can alleviate the shuttle effect and offer good cycling stability.

Reviewer 3 Report

Although this seems like a good idea, the coating of the celgard separator with Lignin does not seem very effective.

The writing is clear, but the scientific argument is poor.

The performance between the LC and the plain separator does not seem very significant. Would be good to see reproducibility.

The EIS is smaller for the LC, but the overpotential losses are larger for the LC sample in comparison to the Celgard sample. Please explain this discrepancy.

The mechanism proposed is that the LC blocks the pores. This seems like a bad idea as it will potentially limit Li ion migration. Maybe this is also the reason for the large overpotential loss.

The explanation that the low Rct for the LC coated separator is due to fast li ion diffusion in the LC separator due to the quinone appears spurious. The authors have already mentioned that these pores are block due to the LC. Beyond this the Rct will be predominately due to the charge transfer in the electrode bulk.

More literature work is required on the background for this idea, more experimental work is required to show that this is a useful coating material or further modification is required to make the coating effective, more analytical work is required to come to clear conclusion. 

Author Response

Although this seems like a good idea, the coating of the celgard separator with Lignin does not seem very effective.

Point 1: The writing is clear, but the scientific argument is poor.

Response 1: The scientific argument is carefully re-checked and revised where necessary.

Point 2: The performance between the LC and the plain separator does not seem very significant. Would be good to see reproducibility.

Response 2: Sorry for this misleading. As shown in Figure 5a, the neat Celgard separator displays a poor cycling performance as 0.33 % decay per cycle (0.15 % decay per cycle during 200 – 500 cycles), while the LC separator presents only 0.06 % decay per cycle during 200 – 500 cycles. Obviously, the LC separator outperforms the neat Celgard separator a lot.

Point 3: The EIS is smaller for the LC, but the overpotential losses are larger for the LC sample in comparison to the Celgard sample. Please explain this discrepancy.

Response 3: This is a good comment. As it can be seen in Figure 5c-d, the LC separator showed larger overpotential losses than the Celgard separator during the first 10 and 100 cycles, which might be due to the coated LNPs inevitably shield part of the pores on neat Celgard and thus resulting in an increased lithium ion diffusion resistance. However, when further cycling to 200 cycles or more, there was unobvious difference noted between these two separators, suggesting the LC separator is as reliable as the commercially available Celgard separator.

Point 4: The mechanism proposed is that the LC blocks the pores. This seems like a bad idea as it will potentially limit Li ion migration. Maybe this is also the reason for the large overpotential loss.

Response 4: Yes, this is a bad idea and the proposed mechanism is inappropriate. Thus, it is deduced that the chemical/physical adsorption between polysulfides and the abundant functional groups/nanostructures of LNPs on the LC separator could mitigate the shuttle effect and thus contributed to the enhanced performance.

Point 5: The explanation that the low Rct for the LC coated separator is due to fast li ion diffusion in the LC separator due to the quinone appears spurious. The authors have already mentioned that these pores are block due to the LC. Beyond this the Rct will be predominately due to the charge transfer in the electrode bulk.

Response 5: As responded above, the abundant functional groups/nanostructures of LNPs on the LC separator are expected to induce strong chemical/physical adsorption of polysulfides and restrict the polysulfides on the cathode, ensuring their better contact than the Celgard separator. Thereby, the LC separator has lower Rct. Moreover, it is found that this benefit of the LC separator would be compromised when the LNPs loading is increased to 3.20 g·cm-2, which confirmed the above speculation because higher LNPs loading usually leads to severer agglomeration (Figure 5e).

Point 6: More literature work is required on the background for this idea, more experimental work is required to show that this is a useful coating material or further modification is required to make the coating effective, more analytical work is required to come to clear conclusion.

Response 6: More related literatures are cited in the Introduction Section, also more experimental details and analytical work are added.

Reviewer 4 Report

Manuscript No: Polymers-577378 by Zhang et al. and titled:
Lignin Nanoparticles-coated Celgard Separator for High-performance Lithium–sulfur Batteries.
This reviewer would like to recommend this manuscript to be published in Polymers after addressing the following issues:
1. The authors need to rewrite the introduction of this manuscript. It is very short. Sevaral methods have been used to modify the PP/PE separators, for example, by carbon nanotubes coatings, integrated structure of sulfur and graphene, conductive mesoporous carbon layer, Nafion coated PP Celagrd separator, Al2O3-coated separator with porous channels and others. See [Advanced Materials, 2015. 27(4): p. 641-647; Journal of Power Sources, 2014. 251: p. 417-422; Journal of Power Sources, 2012. 218: p. 163-167; Electrochimica Acta, 2014. 129: p. 55-61]
2. The authors need to investigate the adhesion of LNPs on the Celgard separator. A peeling test is very common is used to investigate the adhesion of nanoparticles or nanofibers on Celgard separator see [J. of solid state electrochemistry, 18, 2451-2458, 2014 and J. APPL. POLYM. SCI. 2013, DOI: 10.1002/APP.38894]
3. How about the thickness of the coated membrane and its effects on the impedance of the Li-S cell and conductivity?
4. Please combine the results of Figures 2b and 2c in one figure (the TGA scan and DTG profile/derivative/ should be together in one figure)
5. The authors need to show XPS results of the separator (with and without coating) with the fitting curves.
6. The authors need to give the equivalent circuit(s) for Fig 5e.
7. The authors need to show results on the electrolyte uptake in terms of electrolyte capacity versus time for the Celgrad and LC separators
8. This manuscript contains a few English errors and typos and should be revised and proofread by an English language expert. Examples: 1) page 2: LNPs is easily deposited? Thanks to its good dispersity in the solvents? a sustainable and low cost energy?,

Author Response

This reviewer would like to recommend this manuscript to be published in Polymers after addressing the following issues:

Point 1: The authors need to rewrite the introduction of this manuscript. It is very short. Sevaral methods have been used to modify the PP/PE separators, for example, by carbon nanotubes coatings, integrated structure of sulfur and graphene, conductive mesoporous carbon layer, Nafion coated PP Celagrd separator, Al2O3-coated separator with porous channels and others. See [Advanced Materials, 2015. 27(4): p. 641-647; Journal of Power Sources, 2014. 251: p. 417-422; Journal of Power Sources, 2012. 218: p. 163-167; Electrochimica Acta, 2014. 129: p. 55-61].

Response 1: The introduction is rewritten by adding more related literature and the suggested papers.

Point 2: The authors need to investigate the adhesion of LNPs on the Celgard separator. A peeling test is very common is used to investigate the adhesion of nanoparticles or nanofibers on Celgard separator see [J. of solid state electrochemistry, 18, 2451-2458, 2014 and J. APPL. POLYM. SCI. 2013, DOI: 10.1002/APP.38894].

Response 2: The adhesion of LNPs on the Celgard separator was tested by referring the mentioned papers (cited in the section 2.2), which indicates LNPs have good adhesion on the Celgard substrate.

Point 3: How about the thickness of the coated membrane and its effects on the impedance of the Li-S cell and conductivity?

Response 3: The thicknesses of the LC separator and the Celgard separator are 34.9 ± 2.0 μm and 25.1 ± 2.7 μm, respectively, both of which represent the average of five measurements.

Point 4: Please combine the results of Figures 2b and 2c in one figure (the TGA scan and DTG profile/derivative/ should be together in one figure).

Response 4: Figures 2b and 2c are combined as one figure as advised, with which the corresponding statement in the caption is also revised.

Point 5: The authors need to show XPS results of the separator (with and without coating) with the fitting curves.

Response 5: Since the main advantages of LNPs over neat Celgard are extra physical and chemical adsorption rather chemical bonds, XPS are not of urgent need in this study. It is sure a good advice, which does inspire us and will be further studied in the near future.

Point 6: The authors need to give the equivalent circuit(s) for Fig 5e.

Response 6: The equivalent circuit(s) for Fig 5e is added and the new Fig 5e is presented as shown below.

Point 7: The authors need to show results on the electrolyte uptake in terms of electrolyte capacity versus time for the Celgrad and LC separators.

Response 7: The LC separator shows no significant difference from the Celgard separator regarding to the electrolyte uptake in terms of electrolyte capacity versus time (as shown below, both of the LC separator and Celgard separator were placed in an electrolyte solution and maintained for over 24 hrs), which can be understood by the fact that LNPs coated on Celgard have no effect on its porosity (Figure 3 a-d) thus remains the good electrolyte uptake performance of Celgard. This mechanism is well addressed in a reported literature, (Journal of Membrane Science, 2019: 117550), which is cited in the revised manuscript.

Point 8: This manuscript contains a few English errors and typos and should be revised and proofread by an English language expert. Examples: 1) page 2: LNPs is easily deposited? Thanks to its good dispersity in the solvents? a sustainable and low cost energy?

Response 8: The English errors in this manuscript include the mentioned are revised, please see them in red color.

Round 2

Reviewer 2 Report

The authors did a thorough work improving manuscript and it can be published, however, additional english check would be highly beneficial. There are still sentences like one below:

In addition, the electrolyte used in this study was consisted was of 1, 3-dioxolane and 1, 2-dimethoxyethane 1:1 v/v) with bis-(trifluoromethane) sulfonimide lithium (1.0 M) and 1% LiNO3.

Please, look throughout manuscript for such inconsistent language use.

Author Response

Response to Reviewer 2 Comments

Point 1: The authors did a thorough work improving manuscript and it can be published, however, additional english check would be highly beneficial. There are still sentences like one below:

In addition, the electrolyte used in this study was of 1, 3-dioxolane and 1, 2-dimethoxyethane 1:1 v/v) with bis-(trifluoromethane) sulfonimide lithium (1.0 M) and 1% LiNO3.

 Please, look throughout manuscript for such inconsistent language use.

Response 1: The manuscript is carefully re-checked, all revisions are carried out by using the redaction function of WORD. Please see details in below:

“acidic hydrotrope” → “acid hydrotrope”; “thereby significantly suppress the polysulfides shuttle and result in” → “thereby significantly suppressinng the polysulfides shuttle and resulting in”; “12 h to give LNPs” → “12 h to obtain LNPs”; “N2 atmosphere (of 20 mL·min-1)” → “N2 atmosphere of 20 mL·min-1”; “was consisted of” → “was of”; “mutltichannel potentiostat” → “multichannel potentiostat”; “The above findings” → “These results”; “shows unobvious difference from” → “shows unobvious immersion-height difference with respect to”; “24 h later for (e)” → “24 h after (e)”; “Typical SEM images of the Celgard separator (Figures. 3a and b) reveal that there are numerous macropores and mesopores on it” → “SEM images reveal that numerous macropores and mesopores exist in the Celgard separator”; “enabling” → “endowing”.

Reviewer 3 Report

I believe that the Lignin is simply blocking the pores and there is no evidence of the functional group trapping as proposed.

Is the Lignin conductive?

The authors claimed the functional groups of LNPs will be effective in controlling the PS migrations. However, the relevant information to this claim is not found in the manuscript. Also, the FTIR is not well explained and does not potentially shows the presence of effective functional groups.

It is clear from Fig 3 c&d that Celgard is not well coated by the LNPs. Typically, by the vacuum filtration process the celgard PP should be fully laminated by the LNPs as shown conceptually in Fig 1. However, it seems that the LNPs is agglomerated and not uniformly dispersed on the separators. Therefore, the LNPs coated separators still will allow the passage of PS which can be seen in the physical migration test of PSs at 24 hrs in Fig 3f. Even at 0.1 hr, the LNPs coated separator also shows the little diffusion of PS which is not practically feasible.  

The electrolyte up-take experiment is poorly introduced and not labelled.

Why does the pristine Celgard LSB CV only show one oxidations peak? This is not usual for LSB with pristine separator.  Why is there the massive difference between the pristine and the coated LSB? This doesn’t agree with the discharge curves.

LNPs loading was 0.72 g.cm-2 and 3.20 g cm-2?

The sulfur loading is missing. Also, the E/S ratios 23 is quite high. 

The cycling performance is not significantly improved as compared to the pristine separators in the rate performance curve (Fig 4d) – except at 2C. Would be good to discuss.

There is a difference noticed in the cycling performance shown in Fig 5a. Again, the question raises about the sulfur loading? Also, the cycling performance for LNPs is not promising. Would be good to see reproducibility.

The major concern, the overpotential loss for LNPs coated separator is around 500 mV at 10th cycle which is significantly higher as compared to the Celgard PP separator. This observation strongly contradicts the EIS results – which shows that the LC separator should have low potential losses to begin with. 

Author Response

Response to Reviewer 3 Comments

Point 1: I believe that the Lignin is simply blocking the pores and there is no evidence of the functional group trapping as proposed.

Response 1: Sorry for this misleading. It should be due to the functional groups rather than the physical blocking pores.

As carefully addressed in J. Goodenough’s paper (Energy & Environmental Science, 2015, 8(8): 2389-2395.), materials with electron-donating groups could bind lithium polysulfides. In this study, the C=C benzene ring and CH3 on LNPs, of which the presences were confirmed by FTIR shown in Figure 2a, are electron-donating groups. In contrast to the ester group mentioned in J. Goodenough’s paper, the C=C benzene ring and CH3 are much weaker in respect of donating electron, thus explain why the LC separator did not show significant difference over the pristine Celgard separator in Figures 3e and f, though much enhanced cycling stability was observed in Figure 5a.

Accordingly, related discussion is added in section 3.1 as “Note, though weaker than the ester group studied in J. Goodenough’s paper [42], the C=C benzene ring and CH3 on LNPs are still of electron-donating groups and thus are expected to bind polysulfides to alleviate the shuttle effect.”, with which the mentioned paper is also cited as ref 42#.

Point 2: Is the Lignin conductive?

Response 2: The Lignin is not conductive. However, the purpose to coat lignin here was not to add extra active material but to offer extra electron-donating groups to bind lithium polysulfides and alleviate the shuttle effect.

Point 3: The authors claimed the functional groups of LNPs will be effective in controlling the PS migrations. However, the relevant information to this claim is not found in the manuscript. Also, the FTIR is not well explained and does not potentially shows the presence of effective functional groups.

Response 3: This question is responded in detail in Response 1, please see above. Besides, as similar FTIR peaks were addressed in ref 37# and 41#, thus the observation of them LNPs could indicate the presence of effective functional groups here.

Point 4: It is clear from Fig 3 c&d that Celgard is not well coated by the LNPs. Typically, by the vacuum filtration process the celgard PP should be fully laminated by the LNPs as shown conceptually in Fig 1. However, it seems that the LNPs is agglomerated and not uniformly dispersed on the separators. Therefore, the LNPs coated separators still will allow the passage of PS which can be seen in the physical migration test of PSs at 24 hrs in Fig 3f. Even at 0.1 hr, the LNPs coated separator also shows the little diffusion of PS which is not practically feasible.

Response 4: This is a good comment. As responded to point 1, instead of physical blocking pores, the enhanced performance of the LC separator should be due to the functional groups. In detail, the C=C benzene ring and CH3 are much weaker in respect of donating electron, thus explain why the LC separator showed the diffusion of PS as shown in Figure 3e. Even though, after cycling for 500 cycles, and f, the LC separator could still display enhanced cycling stability over the Celgard separator (Figure 5a).

Based on above, all “strong chemical /physical adsorption” are accordingly revised as “chemical binding”.

Point 5: The electrolyte up-take experiment is poorly introduced and not labelled.

Response 5: (1) The caption of Figure 3g is improved as “(g) photograph showing electrolyte immersion-height comparison between both separators after 60 min, of which the dimension was fixed at 1.5 cm × 18 cm”;

(2) the figure is improved with new labels as shown below.

Point 6: Why does the pristine Celgard LSB CV only show one oxidations peak? This is not usual for LSB with pristine separator.  Why is there the massive difference between the pristine and the coated LSB? This doesn’t agree with the discharge curves.

Response 6: (1) The CV of the Celgard LSB was repeated that gives a usual view of that for LSB with pristine separator, which is added as the new Figure 4b (as shown below).

(2) According to a reported literature (Journal of Alloys and Compounds, 2017, 709: 677-685.), two different peaks appeared at ~1.9 V and ~2.5 V of the CV of the LC LSB could be attributed to the further reduction of the Li2Sn to precipitate Li2S2/Li2S and the oxidation of Li2S to elemental sulfur, respectively.

Point 7: LNPs loading was 0.72 g.cm-2 and 3.20 g cm-2?

Response 7: Yes, it was.

Point 8: The sulfur loading is missing. Also, the E/S ratios 23 is quite high.

Response 8: (1) According to the TGA result shown in Figure 4c, the real sulfur loading of the cells used in this study is estimated to be 73.8 wt% or 1.5 mg·cm−2;

(2) To the best of our knowledge, the E/S ratio of 23 is not high. As mentioned in paper (Nature nanotechnology, 2019: 1), an electrolyte content of below 3 ml per 1 g of S (E/S ratio) is required for a competitive sulfur cathode, whereas most studies in the literature use 5–20 ml electrolyte per 1 g S. Also, an E/S ratio of 20 was applied in paper (Advanced Energy Materials, 2018, 8(21): 1800590), E/S ratios of 15 and 30 were applied in paper (Nature Energy, 2018, 3(9): 783), an E/S ratio of ~59 was applied in paper (Nano letters, 2013, 13(12): 5891-5899), E/S ratios of 60 and 78 were applied in paper (ACS Energy Letters, 2017, 2(11): 2591-2597), E/S ratios of 16 and 86 were applied in paper (Journal of Materials Chemistry A, 2014, 2(20): 7383-7388), etc.

Point 9: The cycling performance is not significantly improved as compared to the pristine separators in the rate performance curve (Fig 4d) – except at 2C. Would be good to discuss.

Response 9: As addressed in detail above, the purpose to coat lignin here was not to add extra active material but to offer extra electron-donating groups to bind lithium polysulfides and alleviate the shuttle effect. In other words, the LC separator could improve the cycling stability instead of the rate performance.

Point 10: There is a difference noticed in the cycling performance shown in Fig 5a. Again, the question raises about the sulfur loading? Also, the cycling performance for LNPs is not promising. Would be good to see reproducibility.

Response 10: (1) As responded above, the real sulfur loading of the cells used in this study is estimated to be 73.8 wt% or 1.5 mg·cm−2.

(2) Figures 5a demonstrated that the LC separator possessed much better cycling stability than the Celgard separator at 1 C. Even after 500 cycles, there was still a high enhancement as 74 % observed for the  LC separator over the Celgard separator (as highlighted in the below figure).

Point 11: The major concern, the overpotential loss for LNPs coated separator is around 500 mV at 10th cycle which is significantly higher as compared to the Celgard PP separator. This observation strongly contradicts the EIS results – which shows that the LC separator should have low potential losses to begin with.

Response 11: The galvanostatic charge/discharge measurement of the Celgard LSB was repeated and the related data is graphed as the new Figure 5c (as shown below), which shows similar overpotential loss as the Celgard separator LSB.

Reviewer 4 Report

The authors responded to all the comments raised by this reviewer. This paper can be published in polymers 

Author Response

Response to Reviewer 4 Comments

Point 1: The authors responded to all the comments raised by this reviewer. This paper can be published in polymers.

Response 1: Thank you for the decision, I’ll keep working on this subject.
